# Molecular Phylogenetics of Seven Cyprinidae Distant Hybrid Lineages: Genetic Variation, 2nNCRC Convergent Evolution, and Germplasm Implications

**DOI:** 10.3390/biology14111527

**Published:** 2025-10-30

**Authors:** Ziyi Wang, Yaxian Sun, Ting Liao, Hui Zhong, Qianhong Gu, Kaikun Luo

**Affiliations:** Engineering Research Center of Polyploid Fish Reproduction and Breeding of the State Education, Ministry, College of Life Sciences, Hunan Normal University, Changsha 410081, China; 202130233024@hunnu.edu.cn (Z.W.); 202130233026@hunnu.edu.cn (Y.S.); 19310051403@163.com (T.L.); 202020141287@hunnu.edu.cn (H.Z.)

**Keywords:** Cyprinidae, hybrid lineage, phylogenetic trees, genetic distances, haplotype networks

## Abstract

**Simple Summary:**

Distant hybridization fuels trait innovation and speciation, but the stabilization mechanism of Cyprinidae interspecific distant hybrid lineages remains unclear. We analyzed 7 such hybrid lineages and their parents using 4 mitochondrial genes (*Cytb*, *COI*, *16S rRNA*, *D-loop*) and 5 nuclear genes (*EGR2b*, *IRBP2*, *RAG1*, *RAG2*, *RH2*), with 41 Cyprinidae species (85 samples) from GenBank supplementing the dataset. The hybrids exhibited variation patterns analogous to other Cyprinidae species; Maximum Likelihood (ML) and Bayesian Inference (BI) trees displayed congruent, well-supported topologies. Most hybrids clustered intermediately between their parental species with maternal affinity, except 2nNCRC (derived from distant hybridization between *Cyprinus carpio* and *Megalobrama amblycephala*), which showed convergent evolution toward *Carassius auratus*. These nine genes and the integrated marker system advance research on cytonuclear coadaptation and support parentage tracing, germplasm conservation, and hybrid breeding—laying a foundation for studies on hybrid speciation and the development of elite aquaculture germplasms.

**Abstract:**

Distant hybridization is key to trait innovation and speciation, with Cyprinidae hybrid phylogeny helping to clarify diversification mechanisms. Yet, a major gap persists in Cyprinidae studies: the stabilization mechanisms of interspecific distant hybrid lineages. To address this, we systematically analyzed the molecular phylogeny of seven Cyprinidae distant hybrid lineages and their parental species, using an integrative genetic framework encompassing four mitochondrial genes (*Cytb*, *COI*, *16S rRNA*, *D-loop*) and five nuclear genes (*EGR2b*, *IRBP2*, *RAG1*, *RAG2*, *RH2*). Homologous sequences of 41 representative Cyprinidae species (85 samples) were retrieved from GenBank to supplement the dataset. Phylogenies were reconstructed from concatenated sequences, complemented by haplotype networks. Intra-/interspecific divergence was quantified using two mitochondrial genes (*COI*, *Cytb*) and two nuclear (*RAG1*, *RH2*). The results showed that these hybrid lineages exhibited variation patterns analogous to other Cyprinidae species. Both ML and BI trees reconstructed exhibited congruent topologies with high support (bootstrap/BPP > 80%), resolving genus/species-level relationships. While most hybrids clustered intermediately between their parental species, they typically displayed maternal affinity. A notable exception was the 2nNCRC (a homodiploid hybrid from *Cyprinus carpio* ♀ × *Megalobrama amblycephala* ♂), which displayed convergent evolution toward *Carassius auratus*. COI-based K2P genetic distance analysis revealed 2nNCRC had a much closer relationship with *C. auratus* (0.0119) than with its parents (0.1249 to *C. carpio*, 0.1552 to *M. amblycephala*). These nine genes elucidate the genetic relationships between Cyprinid hybrid lineages and progenitors, serving as pivotal molecular markers for parentage tracing and genetic dissection of distant hybridization mechanisms. The integrated mitochondrial–nuclear marker system in this study advances understanding of cytonuclear coadaptation and the stabilization of interspecific distant hybrid lineages in Cyprinidae. Specifically, it provides a precise tool for parentage tracing, Cyprinid germplasm conservation, and targeted regulation of hybrid breeding—laying a foundation for exploring hybrid speciation and developing elite aquaculture germplasms.

## 1. Introduction

Hybridization and introgression act as key drivers of biological evolution, particularly in rapidly radiating lineages, facilitating speciation and adaptive diversification [1,2,3,4,5,6]. Mounting evidence reveals hybridization as pivotal in sustaining lineage diversification and driving speciation [1,7]. Intergeneric hybridization promotes adaptive trait recombination and the emergence of novel phenotypes via subgenome reconfiguration in teleosts [8,9]. Through laboratory-induced distant hybridization, fertile allotetraploid and autotetraploid fish were produced, providing genomic evidence of hybrid speciation [10,11,12].

Comprising over 37,000 species [13], fish represent the most diverse group of lower vertebrates, thereby serving as a crucial genetic reservoir for biodiversity conservation. Interspecific introgression, a phenomenon widely observed across various fish families such as Cobitidae, Poeciliidae, Atherinidae and Cyprinidae, significantly influences diversification patterns [14,15,16]. Notably, it played a pivotal role in the explosive radiation of African cichlids through genomic recombination [1,17,18]. The role of distant hybridization in fish speciation warrants deeper investigation; however, current insights into its genetic architecture and regulatory networks remain limited and fragmentary, necessitating further validation via advanced technical approaches.

Cyprinidae comprise 1799 valid species in 167 genera [19]. With remarkable species diversity and widespread geographical distribution, they constitute pivotal aquaculture resources in China [20]. Notably, hybridization among *Cyprinus*, *Carassius*, and *Megalobrama* generates evolutionarily significant and economically vital lineages, making research into their distant hybridization highly crucial [5,21,22]. Distant hybridization generates progeny combining advantageous phenotypes, enabling novel aquaculture varieties that provide elite germplasm resources. Concurrently, the unique morphology, phylogenetic affinities, and inheritance architectures of hybrid progeny relative to parental species elucidate genetic principles of fish hybridization—critical for deciphering Cyprinidae’s systematic evolution and mechanisms of species diversification [5,9,23]. Decades of systematic distant hybridization in fish, incorporating self-crossing, backcrossing, and gynogenetic manipulations, have enabled our laboratory to develop multiple hybrid cyprinid lineages—including allotetraploid (4*n* = 200), autotetraploid (4*n* = 200), allodiploid (2*n* = 100), and autodiploid strains [9]. These experimentally stabilized lineages serve as indispensable models for decoding hybridization-driven inheritance patterns in Cyprinidae and pioneering next-generation germplasm for sustainable aquaculture. Specially, these resources uniquely resolve macro-/micro-hybridization dynamics and empower germplasm engineering [23]. Our laboratory has obtained approval for the development of eight new national-level aquaculture varieties (https://ldbff.hunnu.edu.cn/kycg/pxhpz.htm, accessed on 30 September 2023). Over the past few decades, our lab has developed a comprehensive theoretical and technological framework for fish distant hybridization. This includes the establishment of “one-step” and “multi-step” breeding methodologies, as well as gynogenetic manipulation techniques. This foundational work has led us to introduce the innovative concepts of “macro-hybridization” and “micro-hybridization” in fish genetic breeding [8,23]. However, the molecular phylogenetics of novel strains/varieties from distant hybridization and gynogenesis remain understudied. Crucially, emerging evidence implicates hybridization in speciation—as exemplified by the crucian carp (2nNCRC) origin hypothesis involving common carp × blunt snout bream hybridization [5,8]. Thus, systematic molecular phylogenetic frameworks are urgently needed to underpin these innovations.

In this study, we performed comprehensive research on the molecular phylogenetics of hybrid fish lineages and strains within *Cyprinus*, *Carassius*, and *Megalobrama* species. Employing a mito-nuclear DNA barcoding approach—incorporating mitochondrial genes (*16S rRNA*, *COI*, *Cytb*, *D-loop*) and nuclear genes (*EGR2b*, *IRBP2*, *RAG1*, *RAG2*, *RH2*)—we reconstructed the molecular phylogenetic frameworks of distant hybridization lineages and Cyprinidae taxa. By integrating our analytical results with previously reported Cyprinidae DNA barcoding data, we further elucidated the genetic variation characteristics of these hybrid lineages, clarified their taxonomic positions, and delineated their kinship patterns with respective parental species. Thereby, we provide fundamental data on the genetic mechanisms of distant hybridization in Cyprinidae, offering critical theoretical support and insights for the creation of novel germplasms and the study of hybrid speciation.

## 2. Materials and Methods

### 2.1. Statement on Animal Subjects

All experimental procedures complied with relevant guidelines and regulations, and were reported in accordance with the ARRIVE guidelines (https://arriveguidelines.org, accessed on 12 June 2023). Genetic data were primarily sourced from GenBank (Appendix A, Table 1). Experimental fish (distant hybrids and parental lines) were reared under standard pond-culture protocols at the Polyploid Fish Protection Station, Hunan Normal University. Prior to dissection, fish were deeply anesthetized with 100 mg/L MS-222 (Sigma-Aldrich, St. Louis, MO, USA); pterygiophore and muscle tissues were then excised, snap-frozen in liquid nitrogen, and stored at −80 °C. All piscine experimental protocols were approved by the Animal Ethics Committee of the College of Life Sciences, Hunan Normal University (Approval No. 2024-825), and strictly followed national standards in China’s Laboratory Animal Management Principles.

### 2.2. Samples

This study entailed the collection of samples from laboratory-bred, distant hybrid strains of Cyprinid fish in addition to their parental lines (Table 1), including *Cyprinus carpio haematopterus* (koi carp, KOC), *Carassius auratus* (gold fish, GF), *Carassius cuvieri* (white crucian carp, WCC) × *Carassius auratus* red var. (red crucian carp, RCC) [24], triploid *C. auratus* (3*n* = 150, 3N) [25] × RCC, 3N × *Cyprinus carpio* (common carp, COC), RCC × COC, improved *C. cuvieri* (WCC-L) [26], and two homodiploid hybrids (2nNCRC: new crucian carp-like; 2nNCOC: new common carp-like) derived from hybridization of the same parental species *C. carpio* (♀) × *Megalobrama amblycephala* (♂, blunt-snout brea, BSB) [12]. Three individuals from each aforementioned hybrid strain and its parental line were used for genomic DNA extraction and gene amplification, ensuring the reproducibility of subsequent experiments and the reliability of results. It should be noted that *C. carpio* (♀) and *M. amblycephala* (♂) individuals used for genomic DNA extraction are the exact parental fish that produced the 2nNCRC/2nNCOC hybrids (not conspecific substitutes from other strains or populations). Additionally, all genetic comparisons between hybrids and parents were conducted using these direct parent–offspring sample pairs.

### 2.3. DNA Extraction, PCR Amplification, Cloning and Sequencing

Three samples of each distant hybrid and their parents were collected to extract genomic DNA. Total genomic DNA from the pterygiophore was extracted by routine approaches [27]. The highly conserved PCR primers were used to amplify four mitochondrial genes (mtDNA) (*Cytb*, *COI*, *16S rRNA*, *D-loop*) and five nuclear genes (*EGR2b*, *IRBP2*, *RAG1*, *RAG2*, *RH2*) (Table 2).

PCR reactions were performed in a 50 μL volume containing 10–30 ng genomic DNA, 1.5 mM MgCl_2_, 250 μM dNTPs, 0.4 μM of each primer, and 1.25 U Taq polymerase (TaKaRa). Thermal cycling conditions were as follows: initial denaturation at 94 °C for 5 min; 30 cycles of 94 °C denaturation (60 s), 52–58 °C annealing (30–60 s, Table 2), and 72 °C extension (60 s); and a final extension at 72 °C for 10 min. Most PCR products were directly sequenced via Sanger method (Sangon Biotech Co., Ltd., Shanghai, China). For fragments not amenable to direct sequencing, amplicons were cloned into the pMD18-T Vector by TA cloning, followed by plasmid transformation into *E. coli* DH5α and purification. At least three independent clones per product were sequenced with vector-specific primers (primer walking) on an ABI 3730XL sequencer (Applied Biosystems, Carlsbad, CA, USA).

Additionally, 40 representative Cyprinidae species encompassing 81 samples were retrieved from GenBank, yielding in total 257 mtDNA sequences and 328 nuclear gene sequences, species-sample size correspondence is added to Appendix A.

### 2.4. Genetic Variation and Genetic Distance

Sequences were aligned using ClustalW (https://www.genome.jp/tools-bin/clustalw, accessed on 21 May 2024), imported into MEGA 11.0 [35], and analyzed for: Conserved sites (C), Variable sites (V), Singleton sites (S), and Parsimony-informative sites (P), transition/transversion ratio (R) and indels, then analyzed in DNAsp 5.1 to determine the synonymous/nonsynonymous mutation ratio (Ks) per gene [36]. Furthermore, sequence homology and variations among amplified fragments from distant hybrids and their parents were analyzed using BioEdit (v5.0.9) [37] and ClustalW (v1.8). Genetic distances (intra- and interspecific) for Cyprinidae species and distant hybrid lineages were computed from four genes (*COI*, *Cytb*, *RAG1*, *RH2)* using the Kimura 2-parameter (K2P) model in MEGA 11.0.

### 2.5. Phylogenetic and Haplotype Network Analysis

Four mtDNA and five nuclear genes from distant hybrids and Cyprinidae specimens were concatenated separately to build mitogenomic and nuclear phylogenetic trees. Furthermore, a combined mito-nuclear dataset (all nine genes) was used to construct the final phylogenetic tree, given no mito-nuclear discordance. All sequences were aligned using MAFFT (v7.313) within PhyloSuite (v1.2.2) [38]. For mtDNA and nuclear genes (Appendix A), Catostomidae were designated as outgroups—they have a more recent common ancestry with Cyprinidae than other extant Cypriniform lineages (pharyngeal feeding structures [39]), hence *Myxocyprinus asiaticus* (Bleeker, 1864) was used as outgroup. Gene saturation of the nine-gene dataset was assessed in DAMBE v.6.4.41 [40]. The test revealed that all Iss (Index of Substitution Saturation) values were significantly lower than Iss.critical (*p* < 0.0001), confirming the suitability of these genes for phylogenetic reconstruction. Additionally, homogeneity was assessed for the mito-nuclear concatenation (nine genes) using the partition-homogeneity test in PAUP v4.0a [41], yielding no significant heterogeneity (*p* = 0.17 > 0.05).

For phylogenetic reconstruction, we first identified optimal substitution models for each gene fragment via ModelTest 3.7 (https://evomics.org/resources/software/molecular-evolution-software/modeltest/, accessed on 1 July 2024). Model selection was performed in PartitionFinder under the BIC criterion. Table 3 lists AIC and BIC values for the nine genes, supporting GTR + I + Γ4 as the optimal phylogenetic substitution model. Subsequently, mtDNA (4 genes) and nuclear (5 genes) datasets were analyzed separately to infer phylogenies via Maximum Likelihood (ML) and Bayesian methods. Following sequence alignment with MAFFT, we concatenated all 9 genes via Concatenate Sequence. The ML tree was inferred in RAxML 8.0 with 1000 bootstrap replicates to assess branch support [42]. For Bayesian inference (BI), a partitioned analysis was implemented in MrBayes 3.2.7a [43]. Convergence of independent MCMC runs was monitored via Tracer v1.6 (http://beast.bio.ed.ac.uk/Tracer, accessed on 2 July 2024), terminating when the average standard deviation of split frequencies fell below 0.01. After discarding the first 10% of generations as burn-in, consensus trees were rendered in FigTree 1.4.4 (http://tree.bio.ed.ac.uk/software/figtree/, accessed on 20 July 2024).

Furthermore, haplotype networks were built with only the four mitochondrial genes (*16S rRNA*, *COI*, *Cytb*, *D-loop*), not the five nuclear genes (*EGR2b*, *IRBP2*, *RAG1*, *RAG2*, *RH2*), as nuclear sequences have extensive degenerate bases, which hinder accurate haplotype identification and reliable mutational relationship inference. Mitochondrial gene haplotype analysis was performed individually for four genes using DnaSP 5.1. Processed haplotypes underwent species-group assignment in WinArl35 [44], and integrated datasets were imported into PopART-1.7 [45] to reconstruct TCS-based haplotype networks with graphical refinement [46].

## 3. Results

### 3.1. Nucleotide Composition and Variable Sites

Whether distant hybrids’ genetic variation matches non-hybrid Cyprinidae and how hybridization affects genomic stability are key prerequisites for determining if hybridization drives Cyprinidae diversification. Notably, comparative analyses show that distant hybrids shared highly similar nucleotide compositions with non-hybrid Cyprinidae, offering initial genetic congruence evidence for exploring hybridization’s role in shaping biodiversity (Table 4). Mitochondrial genes showed distinct patterns: *COI* and *Cytb* displayed reduced G content with similar T/C/A proportions; *16S rRNA* contained comparable T/C levels, elevated A, but depleted G; the *D-loop* featured similar T/A content, with G < C and both lower than other mtDNA. Among nuclear genes, *EGR2b* demonstrated low T and high C, with equivalent A/G levels. *IRBP2*, *Rag1*, and *RAG2* maintained balanced base distributions, while *RH2* exhibited comparable T/G content but low A and elevated C. *EGR2b*, *RAG2*, and *RH2* showed pronounced C + G bias, while *IRBP2* and *RAG1* had minimal compositional bias (A + T ≈ C + G). All mtDNA exhibited significant A + T bias, with the *D-loop* showing the greatest divergence. Nuclear genes overall displayed C + G dominance, differing markedly from mitochondrial profiles.

All nine genes showed high conservation in the distant hybrid strain. Conserved site proportions were significantly higher than in other Cyprinidae (Table 5), with the contrast most pronounced in *COI* (hybrids: 79.30% vs. other Cyprinidae: 16.31%). Polymorphism Information Content (PIC) and single nucleotide variants (SNVs) were also substantially lower in the hybrid strain.

Across 9 genes, transitions > transversions; nuclear genes had more variation and insertions-deletions (Indels) than mtDNA (Appendix A). BSB and its hybrids had the highest per-gene Indel frequency (Appendix A). Progeny had fewer synonymous than non-synonymous mutations. Ks was ~10× higher for progeny–paternal vs. progeny–maternal, except 3N × RCC (mtDNA) and WR/RCC × COC (nuclear genes). 2nNCRC had lower Ks with wild *C. auratus* than parents (Appendix A, Appendix A), except *16S rRNA* (2nNCRC-COC Ks = 0 vs. 2nNCRC-*C. auratus* 0.0513 ± 0.0008, Table 2, Appendix A); *EGR2b*/*RAG1* Ks were comparable (Appendix A, Appendix A).

### 3.2. Gentic Distance

Based on the *COI* gene, we calculated intraspecific/interspecific genetic distances for distant hybrid parental–offspring lineages (Table 6) and other Cyprinidae (Appendix A). Intraspecific distances were uniformly 0.0000 (Appendix A). Interspecific/inter-strain distances ranged from 0.0000 (WR vs. WCC-L) to 0.1269 (COC vs. WCC-L). Hybrid–maternal distances: 0.0000–0.1249 (COC vs. 2nNCRC); hybrid–paternal: 0.0054–0.1715 (smallest: 3N × RCC vs. RCC). Offspring–maternal distances were < 0.02 (except COC vs. 2nNCRC), smaller than paternal. Notably, the distance between 2nNCRC and wild crucian carp (0.0119) was lower than to its maternal parent COC. For the other three genes (*Cytb*, *RAG1*, and *RH2*), a consistent pattern emerged: intraspecific genetic distances were lower and less dispersed (predominantly clustering near 0.0000), whereas interspecific genetic distances were more dispersed and exhibited high variability (Appendix A). Notably, across all four genes, interspecific genetic differentiation varied substantially among different species pairs—a finding that reflects the complex genetic relationships between distinct species.

Additionally, across all interspecific comparisons, mtDNA genes (*COI*, *Cytb*) consistently exhibited greater genetic distances than nuclear genes (*RAG1*, *RH2*) (Appendix A). This pattern reflects that nuclear genes are evolutionarily more conserved relative to their mitochondrial counterparts.

### 3.3. Phylogenetic Analysis

Both ML and BI trees showed congruent topologies, with consistency across mtDNA, nuclear genes, and concatenated datasets (Appendix A, Figure 1a,b). No significant nuclear–mtDNA topological incongruence existed. Same-genus species formed monophyletic clades (conspecifics as distinct lineages, except *Garra waterloti*; Figure 1a,b), and high nodal support (ML BS > 85%; BI PP > 0.80; Figure 1a,b) validated Cyprinidae’s nuclear–mitochondrial phylogeny.

The phylogeny resolved three major clades; distantly hybridized strains formed no distinct branch, remaining within Cyprinidae (Figure 1a,b). They clustered with *Cyprinus*/*Carassius*, forming a clade with Barbinae. COC/KOC/2nNCOC grouped with *C. carpio* (*Cyprinus* lineage), and other hybrids with *C. auratus* (*Carassius* lineage). 2nNCRC formed a discrete clade with RCC × COC, WCC-L, WCC, WR, and RCC occupying an intermediate phylogenetic position between *C. auratus* and *C. carpio*; 3N × COC/3N × RCC nested in 3N/*C. auratus*/RCC subclade (Figure 1a,b).

Furthermore, phylogenetic clustering consistently grouped hybrid strains with their maternal lineages—e.g., WR with WCC, 3N × RCC/3N × COC with 3N, and RCC × COC with RCC—confirming stronger maternal phylogenetic affinity across these distant hybrid lineages. However, 2nNCRC and 2nNCOC (derived from COC × BSB) exhibited divergent phylogenetic positions: while 2nNCOC clustered with *C. carpio*, 2nNCRC grouped with *C. auratus*, revealing far greater genetic divergence in 2nNCRC than other hybrid progeny. Nevertheless, 2nNCRC still showed closer phylogenetic affinity to maternal COC than to paternal BSB.

### 3.4. Haplotype Network Analysis

In the haplotype networks constructed from four genes, we implemented categorical color coding by genus for cyprinid fishes and specific color assignments for distantly hybridized strains. The analysis revealed consistently low variation among haplotypes sharing the same color-coded taxon (genus or species), with multiple shared haplotypes recurring across all four loci.

The *16S rRNA* network (Figure 2) had 34 haplotypes (6 shared, π = 0.0747). Hap_9 was anomalously shared across 7 genera (*Barbus*, *Parasinilabeo*, *Discogobio*, *Ptychidio*, *Sinocrossocheilus*, *Rectoris*, and *Semilabeo*). Hap_28 was fixed in *Labeo forskalii*/*L. parvus*; Hap_2 (*C. carpio*/COC/KOC/2nNCOC), Hap_3 (*C. auratus*/GF/RCC/3N/2nNCRC/3N × COC/RCC × COC), and Hap_33 (WCC-L/WR/WCC) were in hybrids. The *COI* haplotype network identified 56 haplotypes with a nucleotide diversity (π) of 0.1574, and all shared haplotypes were restricted to within genera (Figure 3). Hap_20 was shared between *L. stolizkae* and *L. rohita*. Among distant hybrid lineages, Hap_5 was fixed in *C. auratus*, 3N, 3N × RCC, and 3N × COC; Hap_3 was shared by *C. carpio* and 2nNCOC; Hap_52 was conserved across WCC-L, WR, and WCC; Hap_53 was present in GF, RCC × COC, and 2nNCRC. *Cytb* networks revealed 59 haplotypes (π = 0.2766; Figure 4) featuring three shared haplotypes absent in *COI*. Hybrid strains showed lineage-specific fixation: Hap_55 (GF/RCC × COC/2nNCRC); Hap_57 (3N/3N × RCC/3N × COC); Hap_54 (WCC-L/WCC/WR). *D-loop* analysis identified 41 haplotypes (π = 0.2141; Figure 5) with 3 shared haplotypes but no intergeneric sharing. Hap_4 was shared among *C. auratus*, 3N, 3N × RCC, and 3N × COC. Hap_38 occurred in RCC × COC, 2nNCRC, and GF. Hap_36 was conserved across WCC-L, WCC, and WR.

## 4. Discussion

Distant hybridization drives speciation and novel variety breeding, necessitating deeper exploration of its genetic basis in cyprinid fishes. This study examined three key aspects: (i) intergenerational genetic variation in cyprinid hybrids, (ii) hybrid–progenitor affinity networks, and (iii) hybrid lineages’ systematic position in Cyprinidae radiation. Results showed that distant hybridization-derived cyprinid lineages retained their phylogenetic placement in Cyprinidae, with closer affinity to progenitors and frequent intermediate positions in phylogenies. Notably, the diploid hybrid 2nNCRC exhibited significantly smaller genetic distance to wild crucian carp than to its parents. These four mtDNA and five nuclear genes thus serve dual roles: as anchors for cyprinid phylogenetics and markers for deciphering hybridization mechanisms relevant to breeding.

### 4.1. Distant Hybridization in Cyprinidae: Mitochondrial Conservation and Nuclear Gene Co-Adaptation

Nucleotide composition and variant site density characterize genetic variation across the nine genes, with marked heterogeneity in base composition and polymorphism (Table 4 and Table 5, Appendix A). Consistent with prior studies [47,48,49], animal mitochondrial genomes show pronounced base composition skewness (A-T/G-C bias). Mitochondrial base composition diverges conservatively: *D-loop* [50] is G-deficient (G = 15.1%) with balanced pyrimidine/purine ratios; *COI*/*Cytb* [51] have low G-content (≤15.5%) and symmetric T/C/A frequencies. Hao et al. (2023) confirmed A + T-enrichment (64.0 ± 3.2) in Leuciscinae, matching Cyprinidae mtDNA skewness [52]. Hybrid mitochondrial genes also exhibit significant A + T enrichment (*COI*:54.7%; *Cytb*:56.8%; *16S rRNA*:57.4%; *D-loop*:65.2%), consistent with accelerated cyprinid mtDNA evolution. Nuclear genes align with prior findings [53,54]: *RAG1* is conserved; *EGR2b* has A/G symmetry and C-bias; IRBP is neutral; *RH2* is A-depleted. All nuclear loci except *IRBP2* exhibit G + C bias, a pattern indicative of lineage-specific selection acting on nucleotide composition. Consistent with this locus-specific variation in nuclear genes, distant hybridization generates progeny with elevated nuclear gene plasticity alongside mtDNA stability; quantification of neutral substitution rates (Ks) and indel burdens has further confirmed that structural variation is more pronounced in nuclear genomes than in mtDNA (Appendix A, Appendix A). Notably, the distant hybrid 2nNCRC and its parents exhibit prominent indel mutations. Except for *16S rRNA*, the Ks value between 2nNCRC and wild Carassius auratus is significantly lower; meanwhile, 2nNCRC’s distinct genetic divergence from both parents directly validates previously reported hybrid-induced genome restructuring [5,12].

Mitochondrial genes (*COI*, *Cytb*, *16S rRNA*, *D-loop*) of hybrid progeny retain marked A + T enrichment, with their base bias aligning closely with that of other Cyprinidae species—a finding that further confirms the stability of maternal inheritance in distant hybridization. This conservatism serves as a “molecular marker” for tracing maternal lineages while also verifying the widespread elimination of paternal mtDNA in distant Cyprinidae hybridization. In cyprinid fish, paternal mtDNA is subjected to developmentally programmed elimination and epigenetic silencing, enforcing uniparental mitochondrial inheritance [55,56,57]. This mechanism ensures clonal transmission of maternal mtDNA haplotypes in hybrids, eradicating heteroplasmic recombination and its associated indel burden. Notably, even during drastic nuclear genome recombination (e.g., in 2nNCRC relative to its parents), the core functional regions of these mitochondrial genes remain stable to maintain energy metabolism. This likely constitutes a key adaptive strategy for natural hybrid progeny to avoid lethality caused by “mitochondrial–nuclear genome incompatibility” [58].

Distant hybridization typically disrupts the co-evolved genomic architecture of parental species, triggering extensive recombination of heterospecific nuclear genomes [21,59]. This recombination promotes DNA repair inaccuracies, consequently increasing indel accumulation in recombinant chromosomes. By contrast with mitochondrial gene conservation, nuclear genes exhibit distinct locus-specific variation. All nuclear genes except *IRBP2* exhibited a G + C bias (49.6–61.9%), forming a “complementary” pattern with the A + T bias of mtDNA. This likely arises from the balance between nuclear genome recombination and selection during hybridization [59]. Nuclear genes with G + C bias (e.g., *EGR2b*, *RAG2*, *RH2*) show greater resistance to recombination-induced DNA structural damage, attributed to the high stability of GC base pairs [60]. This advantage enables their preferential retention in hybrids, serving as the “genetic basis” for adapting to novel ecological niches [5].

This study’s nucleotide composition and variation data not only validate Cyprinidae distant hybridization genetic rules but also uncover, at the molecular level, the “mitochondrial conservation + nuclear gene divergence” coordinated adaptive pattern in fish distant hybridization—providing key insights into natural hybrid species formation mechanisms. In Cyprinidae radiative evolution, hybridization operates not merely as a process of “genetic admixture” but as mito-nuclear genomic co-adaptation—one that underpins the survival and evolution of hybrid lineages.

### 4.2. Cytonuclear Topological Congruence and Genetic Distance: Unique Convergent Evolution of 2nNCRC in Cyprinidae

Genetic distance delineates population divergence, with interspecific values typically 10× higher than intraspecific ones (the “10× rule” [61]); for cyprinids, *COI* intraspecific K2P distances are usually <0.02 [62], validating *COI* for species identification [51,63,64]. In this study, most distant hybrid lineages followed this pattern: intraspecific distances were lower than interspecific ones (Appendix A), and hybrid–maternal distances (0.002 ± 0.003) were far smaller than hybrid–paternal ones (0.103 ± 0.063; Table 6), confirming maternal lineage affinity. Overall, a consistent pattern was observed across the four genes (two mitochondrial: *COI*, *Cytb*; two nuclear: *RAG1*, *RH2*): intraspecific genetic distances were much smaller than interspecific distances. Intraspecific distances were relatively concentrated within a low-value range, reflecting high genetic similarity among conspecific individuals. In contrast, interspecific distances were not only greater in magnitude but also highly dispersed—an observation indicating that different species have undergone varying degrees of genetic differentiation during evolution, leading to more distinct genetic differences. Notably, the diploid hybrid 2nNCRC (from *C. carpio* [COC] × *M. amblycephala* [BSB] [5,12]) deviated: it diverged from parents (COC: 0.123; BSB: 0.155) but clustered with Carassius auratus (0.012; Table 6), indicating convergence to wild crucian carp. A *16S rRNA* haplotype network showed limited variation between 2nNCRC and its maternal COC (consistent with clonal mtDNA transmission [12]), while phylogenetic reconstructions grouped most hybrids (e.g., 3N, 3N × RCC) with *C. auratus* (Bayesian PP = 1.00; Figure 1a; Figure 2). Haplotype divergence analysis further highlighted 2nNCRC’s uniqueness: other hybrids had ≤3 substitutions vs. mothers, but ≥20 vs. fathers/non-parental cyprinids.

The topological congruence between ML and BI trees effectively ruled out cytonuclear phylogenetic discordance (Figure 1), a pattern robustly supported by four mtDNA loci and five nuclear loci. Notably, these nine loci have been previously genetically validated as reliable markers for resolving Cyprinidae phylogenetic relationships [65,66,67] and have also been applied to revise Cypriniformes classification [68,69,70]. Importantly, both trees exhibit highly consistent topologies, which together underpin the precise phylogenetic resolution of distant hybrid lineages within Cyprinidae. From the perspective of clade-level clustering patterns, these two phylogenies further reveal two key evolutionary features of Cyprinidae fish: “generic monophyly” (i.e., species within the same genus form a monophyletic clade) and “phylogenetic affinity between hybrid lineages and their parental taxa” (i.e., hybrid lineages show close evolutionary relationships with their parental species in phylogenetic frameworks). Consequently, our nuclear–mitochondrial congruence (Figure 1) resolved *Cyprinus* clades and *Carassius* complex precisely, validating species boundaries and hybrid classifications. Most hybrids clustered with mothers: WR/WCC-L with maternal WCC, 3N-derived crosses with 3N, and 2nNCOC exclusively with COC (paternal exclusion). In contrast, 2nNCRC formed a clade with *Carassius* (not parents), and variant site/genetic distance analyses confirmed its mito-nuclear divergence from progenitors—pointing to a unique evolutionary trajectory [5]. Furthermore, all four tree topologies (mitochondrial BI/ML-trees, nuclear BI/ML-trees in Appendix A) consistently cluster 2nNCRC/2nNCOC within the *C. auratus* clade with robust support, directly ruling out cytonuclear discordance and reinforcing the reliability of our phylogenetic inferences about the hybrid lineages.

Three lines of evidence link 2nNCRC’s evolution to post-hybrid recombination and directional selection. First, nuclear genome recombination and selective allele retention drive 2nNCRC’s divergence from its parents. The five nuclear loci (*EGR2b*, *IRBP2*, *RAG1*, *RAG2*, *RH2*) in 2nNCRC showed higher insertion/deletion (indel) frequencies than those in COC and BSB—especially in BSB-derived regions—indicating extensive post-hybrid genomic reshuffling. More critically, synonymous substitution rates (Ks) between 2nNCRC and wild *C. auratus* (e.g., *EGR2b*: 0.008 ± 0.001) were far lower than those between 2nNCRC and COC (0.072 ± 0.005; Appendix A), revealing directional selection for *C. auratus*-homologous alleles—likely enhancing 2nNCRC’s adaptation to freshwater environments (e.g., improved nutrient utilization and stress resistance) [5]. Second, 2nNCRC’s divergence involves coordinated convergence of both mtDNA and nuclear genomes toward *C. auratus*. While 2nNCRC inherits mtDNA from its maternal parent (COC), as is typical for cyprinids [12], its mtDNA has diverged significantly from COC (genetic distance = 0.1249) and converged toward *C. auratus* (genetic distance = 0.0119; Table 6)—a pattern consistent with adaptive mutation to ensure compatibility with its converging nuclear genome. This coordination is critical for mito-nuclear coadaptation: processes like oxidative phosphorylation require strict alignment between mtDNA-encoded and nuclear-encoded subunits [12], and unilateral convergence of either genome would lead to lethal incompatibilities. Thus, concurrent shifts in both genomes represent an adaptive solution—avoiding functional conflicts while aligning 2nNCRC with *C. auratus*’ ecological niche. Third, 2nNCRC exhibits morphological and functional convergence with *C. auratus*, most notably in pharyngeal teeth—a trait directly linked to feeding ecology. 2nNCRC shares identical pharyngeal teeth morphology with *C. auratus* (4 compressed teeth per side, adapted for omnivory with a preference for aquatic plants), whereas its parents have distinct dentition: COC has rounded molariform teeth (for snail feeding), and BSB has recurved uncinate teeth (for aquatic plant feeding) [5]. This convergence in a key feeding organ reflects selection for a *C. auratus*-like niche, which reduces competition with parental species (COC/BSB) and stabilizes the 2nNCRC lineage.

This evolutionary pattern of 2nNCRC is taxonomically unique among cyprinid hybrids: it is shaped not only by post-hybrid genomic recombination and directional selection but also by lineage-specific mito-nuclear coadaptation and niche-driven functional convergence—supported by both our data and prior research [5,12]. This uniqueness is particularly striking when compared to prior observations of distant cyprinid hybridization, where post-hybrid lineages often face lethal mito-nuclear incompatibilities or genomic instability [8]; 2nNCRC, by contrast, has overcome these bottlenecks. Integrating our current data with earlier findings [5], we conclude that the convergent evolution of both mtDNA and nuclear genomes toward *C. auratus* explains 2nNCRC’s distinct phylogenetic position: it bridges the lineages of its parental species while forming a stable, self-sustaining lineage. However, we acknowledged that the evolutionary mechanisms of 2nNCRC remain partially unresolved. To address remaining knowledge gaps, future studies should (1) identify selection sweeps driving *C. auratus* allele retention; (2) validate niche differentiation via long-term sympatric surveys; (3) test if similar mito-nuclear patterns exist in other cyprinid hybrids.

### 4.3. Align Cyprinidae Hybrid Core Traits: Stabilize Lineages

Our findings align with the core genetic hallmarks of natural hybridization in Cyprinidae, which are primarily defined by two interrelated features: maternal inheritance dominance and genomic stability maintenance. A critical mechanism underpinning these traits is the conserved process of “paternal mitochondrial programmed elimination and epigenetic silencing,” observed across hybridization events within Cyprinidae species [71,72]. These dual mechanisms—maternally biased genetic transmission and paternal mitochondrial clearance/silencing—act synergistically to sustain cellular energy metabolic homeostasis and genomic integrity [55,56,57,73]. Consequently, hybrid offspring predominantly inherit mitochondrial genes from the maternal genotype, while nuclear genes undergo biparental recombination. Notably, the patterns of nucleotide base composition bias detected in the hybrid lineages of this study show high congruence with those of naturally occurring Cyprinidae species (Table 4), reinforcing the genetic continuity between experimental and natural hybrid systems. However, natural hybridization in Cyprinidae is largely constrained to congeneric species or closely related taxa within the same genus [14,15,16], with intergeneric distant hybridization being exceedingly rare. Critically, even the limited number of naturally occurring intergeneric hybrids fail to establish stable lineages due to inherent genomic instability [12], representing a longstanding challenge in evolutionary and breeding research.

Against this background, our study represents a pivotal breakthrough: by employing a “multi-step breeding combined with gynogenesis regulation” strategy [8,12,23], we successfully generated two stable intergeneric hybrid lineages (2nNCRC and 2nNCOC) through distant hybridization between *C. carpio* and *M. amblycephala* [8]. These lineages exhibit remarkable genomic stability, as evidenced by low mutation loads across all nine analyzed genes—including a synonymous substitution rate (Ks) divergence from the maternal *C. carpio* of <0.02 (Appendix A) and Indel frequencies ranging merely from 1.08% to 7.11% (Table 5). This achievement not only resolves the longstanding research gap regarding the stabilization of intergeneric distant hybridization in Cyprinidae but also validates that laboratory-based regulatory methods can effectively mitigate genomic conflicts inherent to natural hybridization. Moreover, it provides critical insights into the mechanisms underlying the formation of “rare stable hybrids” observed in natural ecosystems, bridging theoretical understanding with applied breeding innovation.

### 4.4. Implications for Germplasm Resource Management in Hybrid Breeding

Cyprinidae is a critical strategic germplasm resource for freshwater fisheries in China. However, the widespread use of hybrid breeding has led to increasingly significant genetic introgression, potentially threatening the genetic purity of Cyprinid species [74,75,76]. This introgression not only reduces the inherent genetic diversity of these species and disrupts the ecological balance of natural aquatic systems, but also presents risks to the aquaculture industry, which relies on purebred broodstock [77,78]. To address this issue, a comprehensive prevention and control system that combines molecular monitoring, physical isolation, and ecological regulation must be established to protect Cyprinid germplasm resources and maintain the ecological equilibrium of aquatic environments. The integrated molecular marker system developed in this study, which incorporates four mitochondrial genes and five nuclear genes, facilitates the precise detection of genetic introgression signals in hybrid individuals. This provides an efficient and reliable technical tool for monitoring the germplasm purity of Cyprinid species. Moreover, the study offers valuable insights for optimizing distant hybridization breeding techniques in Cyprinids. It confirms that the mitochondrial genes of hybrid progeny are highly conserved relative to those of the maternal parent. This feature can be leveraged by breeders to prioritize maternal broodstock with favorable mitochondrial characteristics, such as those linked to high growth rates and robust stress resistance. This strategy would facilitate efficient transmission of target traits while minimizing genetic uncertainty in hybrid progeny. Notably, the 2nNCRC hybrid lineage generated in this research exhibits genetic stability. However, it presents a relatively minor genetic distance from wild *Carassius auratus* (genetic distance: 0.0119; Table 6). This implies that if the hybrid were to inadvertently escape into natural aquatic environments, it could potentially integrate into local *Carassius auratus* populations through convergent evolution. This would consequently risk genetic contamination of wild populations. Therefore, it is imperative to implement a rigorous “closed breeding system” and strictly prohibit the release of intergeneric hybrids into natural ecosystems. As such, the molecular marker system developed in this study offers a readily applicable scientific tool for conserving Cyprinid germplasm resources, regulating hybrid breeding processes, and dynamically managing natural populations.

## 5. Conclusions

This study not only confirms the core genetic principles of natural hybridization in Cyprinidae—including dominant maternal inheritance and maintained genomic stability—but also supplements crucial evidence for the stabilization of distant intergeneric hybrid lineages within this family. The integrated molecular marker system (4 mitochondrial + 5 nuclear genes) established herein advances theoretical insights into the evolutionary biology of fish hybridization, while offering practical value across multiple domains: it supports Cyprinidae aquaculture advancement, strengthens biodiversity conservation, enables precise regulation of hybrid breeding, and facilitates evidence-based dynamic management of natural Cyprinid populations. To further clarify hybridization’s role in fish adaptive radiation, future research should (1) decode molecular mechanisms (e.g., locus-specific selective pressure, mito-nuclear coadaptation) driving the convergent evolution of the 2nNCRC hybrid lineage; and (2) expand this marker system to more Cyprinidae hybrid lineages, verifying its generalizability and uncovering lineage-specific patterns of genetic inheritance and divergence in distant hybridization.

## Figures and Tables

**Figure 1 biology-14-01527-f001:**
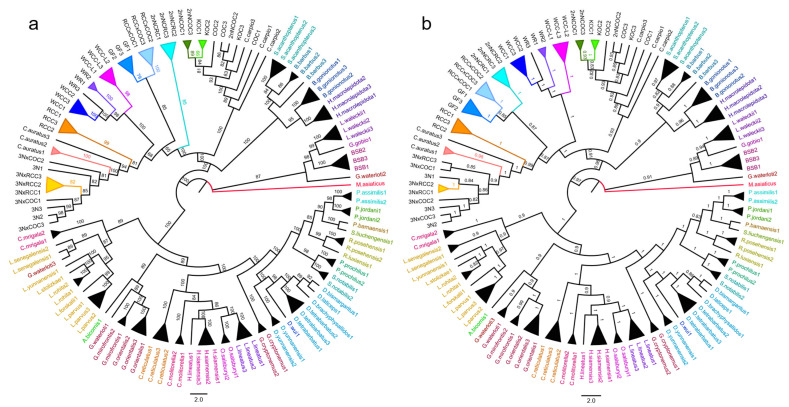
Maximum likelihood (**a**) and Bayesian inference (**b**) phylogenies reconstructed from a concatenated nuclear–mitochondrial gene dataset. Both trees display a highly consistent topology, which collectively underpins the phylogenetic resolution of distant hybrid lineages within Cyprinidae. From the perspective of clade clustering patterns, both phylogenies clearly exhibit two key features of Cyprinidae fishes: “generic monophyly” and “phylogenetic affinity between hybrid lineages and their parental taxa”. In each tree, the names of conspecific cyprinid species within the same genus are uniformly color-coded. In contrast, the names of hybrid lineages and their parental species are marked in black; additionally, different colors of the clustered clades correspond to distinct strains or parental taxa.

**Figure 2 biology-14-01527-f002:**
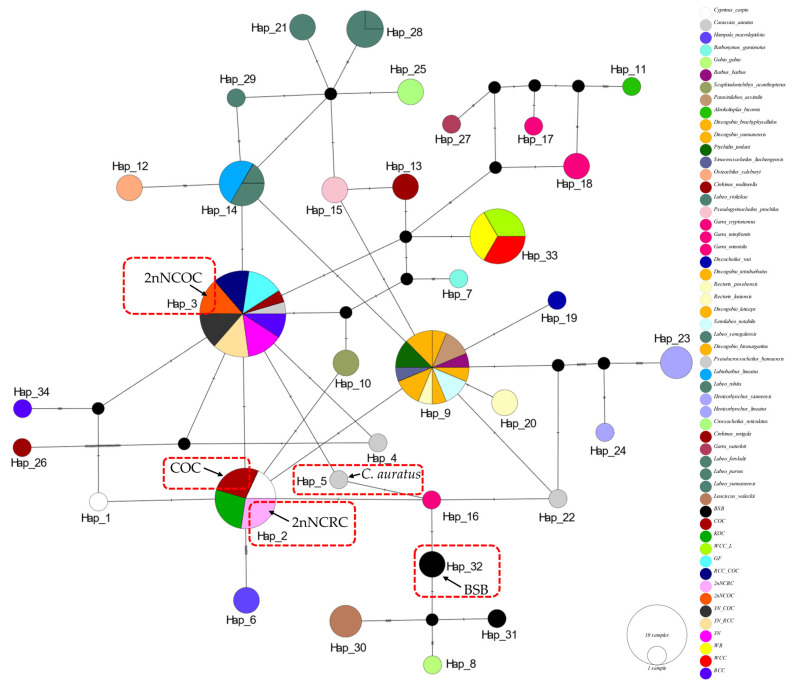
Median-joining network (PopART v1.7) derived from 111 sequences of *16S rRNA* (Appendix A) alignments. Conspecific cyprinid species within the same genus are uniformly color-coded. The haplotypes of two homodiploid hybrids (2nNCRC, 2nNCOC) and their parents (COC and BSB), as well as the *C.auratus* are indicated in the figure. The size of the circles represents haplotype frequency. Each connecting line represents a single nucleotide substitution, and each little short line represents mutated position. The same as below.

**Figure 3 biology-14-01527-f003:**
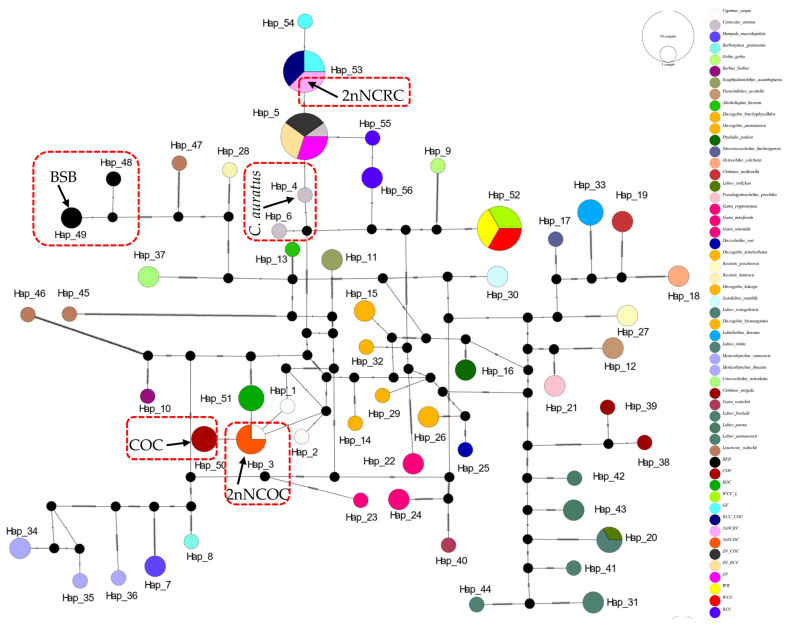
Median-joining network (PopART v1.7) derived from 112 sequences of *COI* (Appendix A) alignments. Conspecific cyprinid species within the same genus are uniformly color-coded.

**Figure 4 biology-14-01527-f004:**
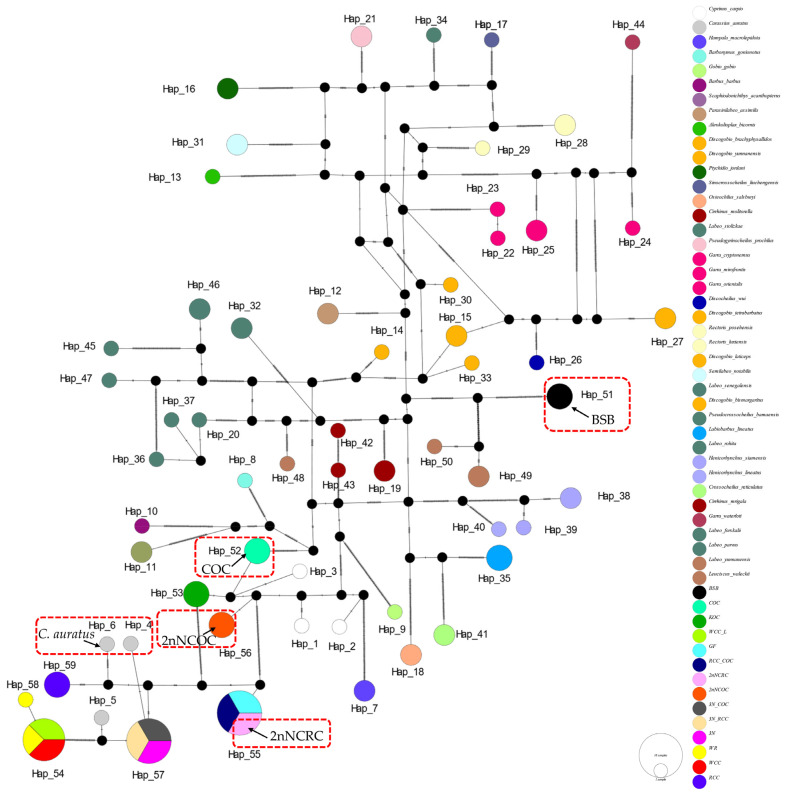
Median-joining network (PopART v1.7) derived from 113 sequences of *Cytb* (Appendix A) alignments. Conspecific cyprinid species within the same genus are uniformly color-coded.

**Figure 5 biology-14-01527-f005:**
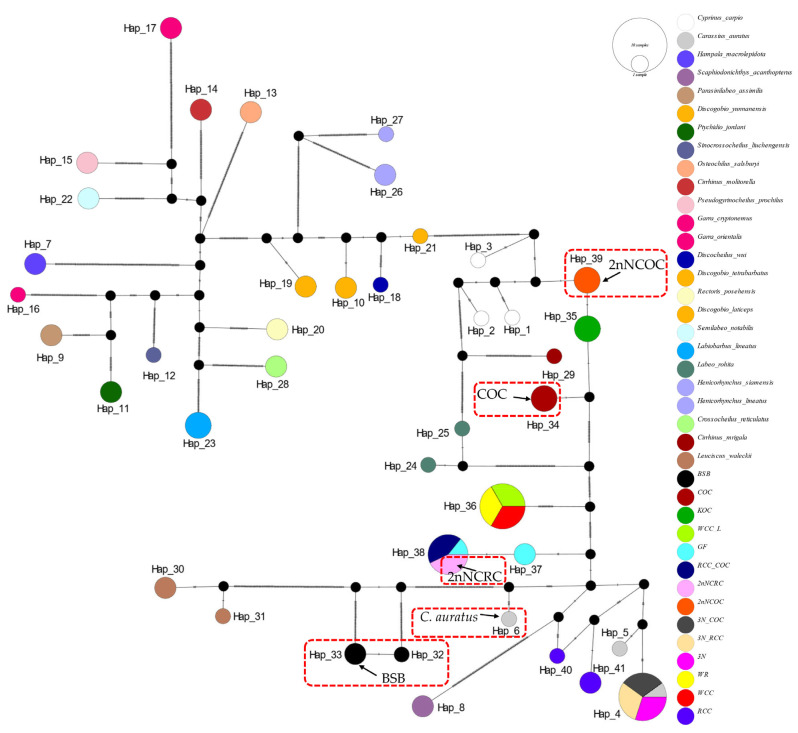
Median-joining network (PopART v1.7) derived from 111 sequences of *D-loop* (Appendix A) alignments. Conspecific cyprinid species within the same genus are uniformly color-coded.

**Table 1 biology-14-01527-t001:** The laboratory-bred strains and seven distant hybrid strains of Cyprinid fishes and their parents.

Species/Strains	Abbreviation	Female Parent	Male Parent
*Cyprinus carpio haematopterus*	KOC	KOC	KOC
gold fish (*Carassius auratus*)	GF	GF	GF
*Carassius cuvieri* × *Carassius auratus* red var.	WR	WCC	RCC
triploid *C. auratus* × *Carassius auratus* red var.	3N × RCC	3N	RCC
triploid *C. auratus* × *C. carpio*	3N × COC	3N	COC
*Carassius auratus* red var. × *C. carpio*	RCC × COC	RCC	COC
new common carp-like homodiploid fish	2nNCOC	COC	BSB
new crucian carp-like homodiploid fish	2nNCRC	COC	BSB
improved *Carassius cuvieri*	WCC-L	WCC	BSB

KOC, *Cyprinus carpio haematopterus*; GF, gold fish (*Carassius auratus*); WR, white crucian carp × red crucian carp; WCC, white crucian carp; RCC, red crucian carp; 3N, triploid *Carassius auratus*; COC, common carp; BSB, M. *amblycephala*; WCC-L, improved *Carassius cuvieri*; 2nNCOC, new common carp-like homodiploid fish (2*n* = 100); 2nNCRC, new crucian carp-like homodiploid fish (2*n* = 100).

**Table 2 biology-14-01527-t002:** Forward and reverse primers for amplifying genes.

Genes	Forward Primers	Reverse Primers	Annealing (°C)	Time (s)
*Cytb* [28]	Glu-F, 5′-CACGARACRGGRTCNAAYAA-3′	Thr-R 5′-ACCTCCRATCTYCGGATTACA-3′	55	30
*COI* [29]	FishF1,5′-TCAACCAACCACAAAGACATTGGCAC-3′	FishR2, 5′-ACTTCAGGGTGACCGAAGAATCAGAA-3′	55	30
*16S rRNA* [30]	16SarL, 5′-CGCCTGTTTATCAAAAACAT-3′	16SbrH, 5′-CCGGTCTGAACTCAGATCACG-3′	55	30
*D-loop* [31]	MitDl-F, 5′-CACCCYTRRCTCCCAAAGCYA-3′	MitDl-R, 5′- GGTGCGGRKACTTGCATGTRTAA-3′	56	30
*RAG1* [32]	RAG1_F, 5′-CTGAGCTGCAGTCAGTACCATAAGATGT-3′	RAG1_R, 5′-TGAGCCTCCATGAACTTCTG AAGRTAYTT-3′	53	40
*RAG2* [33]	RAG2-f2, 5′-ARACGCTCMTGTCCMACTGG-3′	RAG2-R6, 5′-TGRTCCARGCAGAAGTACTTG-3′	58	30
*EGR2b* [34]	E2B287Fd, 5′-TTGACTCSCAGTATCCAGGTAAC-3′	E2B1117Rb, 5′-AGGTGGATTTTGGTGTGTCTYTT-3′	52	60
*IRBP2* [32]	IRBP2_F, 5′-AACTACTGCTCRCCAGAAAARC-3′	IRBP2_R, 5′-GGAAATGCATAGTTGTCTGCAA-3′	55	30
*RH2* [32]	RH_F, 5′-CTAATCCAGATCCTAACTTGCAAAG-3′	RH_R, 5′-CAGTCCAGAGACGTCCGGCGTGGTCT-3′	58	40

**Table 3 biology-14-01527-t003:** The optimal nucleotide substitution model is supported by nine genes, with AIC, AICc, and BIC values calculated accordingly.

Genes	Model	lnL	AIC	BIC
*COI*	TIM + I + G	18,866.729	37,749.457	37,771.133
*16S rRNA*	GTR + I + G	12,142.850	24,305.699	24,332.703
*Cytb*	GTR + I + G	14,400.396	28,820.793	28,841.537
*D-loop*	HKY + G	15,332.923	30,675.859	30,688.414
*EGR2B*	HKY + I + G	6834.516	13,681.033	13,697.397
*IRBP2*	TrNef + I + G	7511.037	15,030.073	15,040.494
*RH2*	HKY + I + G	6834.517	13,681.033	13,697.398
*RAG1*	SYM + I + G	11,766.768	23,547.535	23,567.047
*RAG2*	TVMef + I + G	10,893.170	21,796.318	21,813.203

**Table 4 biology-14-01527-t004:** Nucleotide composition of mtDNA and nuclear genes.

Genes		Total (bp)	Average Content (%)					
			T	C	A	G	A + T	G + C
Distant hybrids	*COI*	681	28.5	28.0	26.2	17.3	54.7	45.3
	*Cytb*	1148	28.2	28.7	28.6	14.6	56.8	43.3
	*16S rRNA*	1147	20.1	23.0	37.3	19.6	57.4	42.6
	*D-loop*	1000	32.7	20.5	32.5	14.3	65.2	34.8
	*EGR2b*	830	17.1	38.4	21.1	23.5	38.2	61.9
	*IRBP2*	872	24.4	24.4	27.6	23.5	52.0	47.9
	*RAG1*	930	25.6	22.7	24.9	26.9	50.5	49.6
	*RAG2*	1055	23.4	26.4	24.1	26.1	47.5	52.5
	*RH2*	872	26.9	30.8	17.9	24.4	44.8	55.2
Cyprinidae	*COI*	681	29.3	25.7	27.6	17.4	56.9	43.1
	*Cytb*	1148	28.3	27.8	29.9	14.0	58.2	41.8
	*16S rRNA*	1147	20.2	23.3	36.7	19.8	56.9	43.1
	*D-loop*	1000	32.7	20.1	34.2	13.0	66.9	33.1
	*EGR2b*	830	17.3	39.0	20.6	23.1	37.9	62.1
	*IRBP2*	872	24.4	24.7	27.4	23.5	51.8	48.2
	*RAG1*	930	24.0	24.3	25.5	26.2	49.5	50.5
	*Rag2*	1055	24.1	27.0	23.4	25.4	47.5	52.4
	*RH2*	872	24.4	33.0	16.9	25.7	41.3	58.7

**Table 5 biology-14-01527-t005:** Variable site information in mtDNA and nuclear genes.

Genes		Conserved Sites (C, %)	Variable Sites (V, %)	Parsimony Informative Sites (PIC, %)	Single Nucleotide Variant (SNV, %)
Distant hybrids	*COI*	540 (79.30%)	140 (20.56%)	138 (20.26%)	2 (0.29%)
	*Cytb*	866 (75.44%)	279 (24.3%)	269 (23.43%)	10 (0.87%)
	*16S rRNA*	999 (87.10%)	136 (11.86%)	127 (11.07%)	9 (0.87%)
	*D-loop*	689 (68.90%)	275 (27.50%)	273 (27.30%)	1 (0.10%)
	*EGR2b*	755 (90.96%)	75 (9.04%)	66 (7.95%)	9 (1.08%)
	*IRBP2*	686 (78.67%)	175 (20.07%)	165 (18.92%)	10 (1.15%)
	*RAG1*	802 (86.24%)	120 (12.9%)	106 (11.4%)	14 (1.51%)
	*RAG2*	783 (74.22%)	267 (25.31%)	191 (18.1%)	75 (7.11%)
	*RH2*	713 (81.77%)	156 (17.89%)	144 (16.51%)	12 (1.38%)
Cyprinidae	*COI*	303 (16.31%)	1265 (68.08%)	643 (34.61%)	616 (33.15%)
	*Cytb*	611 (53.50%)	531 (46.50%)	495 (43.35%)	35 (3.06%)
	*16S rRNA*	1094 (62.41%)	638 (36.39%)	540 (30.80%)	98 (5.59%)
	*D-loop*	966 (50.00%)	959 (49.64%)	915 (47.36%)	44 (2.28%)
	*EGR2b*	653 (78.86%)	175 (21.14%)	131 (15.82%)	44 (5.31%)
	*IRBP2*	488 (57.48%)	306 (36.04%)	300 (35.34%)	61 (7.18%)
	*RAG1*	997 (63.30%)	531 (33.71%)	415 (26.35%)	116 (7.37%)
	*RAG2*	776 (60.44%)	505 (39.33%)	332 (25.86%)	173 (13.47%)
	*RH2*	620 (70.06%)	253 (28.59%)	215 (24.29%)	38 (4.29%)

**Table 6 biology-14-01527-t006:** Calculating genetic distance in *COI* sequences between distant hybrids and their parents based on the Kimura 2-Parameter model. KOC, *Cyprinus carpio haematopterus*; GF, gold fish (*Carassius auratus*); WR, white crucian carp × red crucian carp; WCC, white crucian carp; RCC, red crucian carp; 3N, triploid *Carassius auratus*; COC, common carp; BSB, *M. amblycephala*; WCC-L, improved *Carassius cuvieri*; 2nNCOC, new common carp-like homodiploid fish (2*n* = 100); 2nNCRC, new crucian carp-like homodiploid fish (2*n* = 100). The *COI* gene sequences of *C. auratus* were retrieved from GenBank (Appendix A). In contrast, *COI* gene sequences of all other taxa included in this study were obtained via PCR amplification using genomic DNA extracted from laboratory-bred strains.

	*C. auratus*	BSB	COC	KOC	WCC-L	GF	RCC × COC	2nNCRC	2nNCOC	3N × COC	3N × RCC	3N	WR	WCC	RCC
*C. auratus*	0														
BSB	0.1602	0													
COC	0.1209	0.1481	0												
KOC	0.1185	0.1462	0.0029	0											
WCC-L	0.0514	0.1715	0.1269	0.1232	0										
GF	0.0124	0.1558	0.1255	0.1255	0.0561	0.001									
RCC × COC	0.0119	0.1552	0.1249	0.1249	0.0556	0.0005	0								
2nNCRC	0.0119	0.1552	0.1249	0.1249	0.0556	0.0005	0	0							
2nNCOC	0.1191	0.1462	0.0015	0.0015	0.125	0.1236	0.1231	0.1231	0						
3N × COC	0.0049	0.1584	0.1197	0.1197	0.0492	0.0094	0.0089	0.0089	0.1179	0					
3N × RCC	0.0049	0.1584	0.1197	0.1197	0.0492	0.0094	0.0089	0.0089	0.1179	0	0				
3N	0.0049	0.1584	0.1197	0.1197	0.0492	0.0094	0.0089	0.0089	0.1179	0	0	0			
WR	0.0514	0.1715	0.1269	0.1232	0	0.0561	0.0556	0.0556	0.125	0.0492	0.0492	0.0492	0		
WCC	0.0519	0.1721	0.1274	0.1238	0.0005	0.0566	0.0561	0.0561	0.1256	0.0497	0.0497	0.0497	0.0005	0	
RCC	0.0074	0.1584	0.1191	0.1191	0.0497	0.0099	0.0094	0.0094	0.1173	0.0054	0.0054	0.0054	0.0497	0.0503	0

## Data Availability

The data are contained within the article, and all genes can be obtained by contacting the author (gqh@hunnu.edu.cn).

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
