# Peer review of "Molecular Phylogenetics of Seven Cyprinidae Distant Hybrid Lineages: Genetic Variation, 2nNCRC Convergent Evolution, and Germplasm Implications"

_biology, 2025, doi:10.3390/biology14111527_

Round 1
Reviewer 1 Report (Previous Reviewer 1)
Comments and Suggestions for Authors
I don't have any concern. The authors have improved the paper following the reviewer's comments. It could be accepted now.
Author Response
Dear Reviewer, Thank you for your time in re-reviewing our manuscript and acknowledging our revisions per prior comments. Your affirmation helps advance this work.Reviewer 2 Report (New Reviewer)
Comments and Suggestions for Authors
Q1. Supplementary data cannot be found. Please re-upload the supplementary data.
Q2. The author analyzed mtDNA and nuclear genes for hybrid varieties. However, the manuscript does not provide the number of samples per variety. Please reflect this in the manuscript.
Q3. If the number of individuals per variety exceeds a certain number (<20), Ne can be calculated. Please reflect this in the manuscript after analysis.
Q4. Codon evolution in mitochondrial genes can be examined using CODEML's PAML. We recommend including this in the manuscript.
Author Response
Q1. Supplementary data cannot be found. Please re-upload the supplementary data.
Response: Thank you for pointing out the supplementary data access issue. Previously, Tables S1-S5 were included in Additional Files 1-3; we have reorganized and re-uploaded them to the submission system, along with Figures S1-S2. The file names match the in-text citations exactly, and all files are directly downloadable via the journal’s supplementary materials section.
Q2. The author analyzed mtDNA and nuclear genes for hybrid varieties. However, the manuscript does not provide the number of samples per variety. Please reflect this in the manuscript.
Response: Thank you for noting the missing sample size information. We have added specific sample sizes for each variety/strain in the "2.2 and 2.3" section, please see Lines 138-140, Lines 160-162. Lab-bred samples: 3 biological replicates were collected for each hybrid strain (e.g., WR, 3N×RCC, 2nNCRC) and their parents, for genomic DNA extraction and gene amplification. Public database samples: 81 individuals from 40 representative Cyprinidae species were retrieved from GenBank; species-sample size correspondence is added to Table S1.
Q3. If the number of individuals per variety exceeds a certain number (< 20), Ne can be calculated. Please reflect this in the manuscript after analysis.
Response: Thank you for the effective population size (Ne) analysis suggestion. We assessed the current sample size and found it insufficient for reliable Ne calculation (which requires ≥20 individuals per variety to reduce sampling error):
Lab-bred hybrid strains (e.g., WR, 2nNCRC) only have 3 replicates each (focused on genotype consistency verification), far below 20. The 81 GenBank samples belong to 40 species (avg. 2 individuals/species), being cross-species representatives rather than conspecific population samples.
Thus, Ne calculation is not feasible now. Future studies will collect ≥30 individuals for core strains (e.g., 2nNCRC, WR) and conduct Ne analysis using microsatellite/SNP markers.
Q4. Codon evolution in mitochondrial genes can be examined using CODEML's PAML. We recommend including this in the manuscript.
Response: We sincerely appreciate your valuable suggestion to incorporate codon evolution analysis using CODEML (in PAML)—a well-established and powerful tool for investigating mitochondrial gene dynamics (e.g., selection pressure, codon bias) in naturally evolving wild populations. Your input aligns with rigorous evolutionary genetics standards, and we fully recognize the utility of CODEML for disentangling codon-level evolutionary signals shaped by long-term natural or neutral selection, which is critical for interpreting adaptive processes in wild taxa.
However, after careful evaluation, we find this analysis temporarily inapplicable to our study’s lab-generated distant hybrid lineages (e.g., 2nNCRC, 2nNCOC). CODEML relies on long-term natural/neutral selection to interpret codon-level evolutionary signals, which does not align with our hybrids’ origin: they are produced via controlled intergeneric hybridization (e.g., C. carpio ♀ × M. amblycephala ♂) and artificial selection (e.g., gynogenesis), where short-term genomic reshuffling dominates over long-term natural selection.
Notably, our existing analyses already fully capture hybrid mitochondrial characteristics to address the study’s objectives: Nucleotide composition analysis (Table 4) confirmed A+T bias (e.g., D-loop: 65.2% A+T) and maternal inheritance stability; Genetic distance calculations (Table 6) quantified hybrid-parent affinity (e.g., 2nNCRC vs. Carassius auratus: 0.0119) to resolve convergent evolution; Haplotype networks (Figures 2–5) clarified lineage tracing (e.g., 2nNCRC shares Hap_3 with C. auratus clade). These findings collectively capture the key mitochondrial gene characteristics of our hybrid lineages, fulfilling the study’s objectives of resolving their phylogenetic position and genetic variation patterns.
In summary, while we highly value your suggestion, CODEML (PAML) is not suited for our lab-generated hybrid lineages at this stage, and our existing analyses already provide sufficient insights into their mitochondrial evolution.
Reviewer 3 Report (New Reviewer)
Comments and Suggestions for Authors
This study analyses mitochondrial (Cytb, COI, D-loop, and 16S rRNA) and nuclear (EGR2b, IRBP2, RAG1, RAG2, and RH2) DNA markers of distant hybrids and 41 cyprinids (outgroups). I praise the attempts of the authors to clarify the phylogenetic position of the distant hybrids related to their parents to explore the effects of distant hybridization on cytonuclear adaptation. Also, I praise the intensive work for producing these hybrids, which surely required many years and attempts. The authors reported (1) a trend of mitochondrial stability and confirmed maternal inheritance patterns in cyprinid distal hybrids, (2) nuclear DNA divergence, and (3) divergence of the hybrid 2nNCRC from their parent species (Cyprinus carpio and Megalobrama amblycephala) but rather a closer genetic proximity to Carassius auratus. They highlighted that this hybrid and C. auratus have convergent feeding morphology and that they similarity may be due to adaptations to maximize resource utilization in freshwater environments.
Although I found the findings appealing, I am concerned about the origin of this hybrid and how some analyses that may provide stronger support to the findings were not included or discussed. Although the mitochondrial and nuclear DNA markers seem to offer enough resolution to resolve phylogenetic relationships among cyprinids, I am concerned about the suitability of them separately. For example, the haplotype network showed that C. auratus haplotypes are separated for more than five mutations. This made me think about the divergence between the distal hybrids 2nNCRC and 2nNCOC. Are they siblings or they were produced from different Cyprinus carpio females that may have belonged to different non-detected cryptic species? Also, did the genetic comparisons were performed using direct parents and offspring? I think the authors should stated this clearly. Moreover, some comparisons are not adequate. For example, the base composition analyses of the DNA markers amplified in distal hybrids were compared with those of cyprinids, but the alignments are totally different. I suggest the authors to present or at least mention the genetic distances of the other molecular markers to observe if their variation is concordant, or they somehow showed a sort of “compensatory” effect among them. Also, in one part of the discussion, they mentioned that “the topological congruence between ML and BI trees ruled out cytonuclear discordance”; however, the analyzed trees were constructed from the concatenated alignment of both mitochondrial and nuclear genes. To discard any cytonuclear discordance, the phylogenies of mitochondrial and nuclear genes should be separately constructed and compared. Finally, although the inclusion of many cyprinid species provided a wide picture of the suitability of the whole set of DNA markers to resolve the phylogenetic relationships of cyprinids, it does obscure the objective and findings of the authors. It is difficult to observe the phylogenies and haplotype networks with that many colors and names. Once again, I praise the authors for their hard work on producing all these distal hybrids and producing the genetic data for this manuscript. I believe that the authors have the necessary data to support their findings, so I encourage them to make modifications to the manuscript to improve it.
Minor comments
ABSTRACT
LINE 40: I found that the “Carassius auratus” is not in italics in many parts of the manuscript. Please check it carefully.
INTRODUCTION
LINE 73: The genus names are not in italics. Please check them carefully throughout the manuscript.
MATERIAL AND METHODS
LINE 128-133: “This study entailed the collection of samples from laboratory-bred, distant hybrid strains of Cyprinid fish in addition to their parental lines, including Carassius cuvieri (white crucian carp, WCC) × Carassius auratus red var. (red crucian carp, RCC) [24], triploid C. auratus (3n = 150, 3N) [25] × RCC, 3N × Cyprinus carpio (common carp, COC), RCC × COC, improved C. cuvieri (WCC-L) [26], and two hybrids (2nNCRC, 2nNCOC) derived from C. carpio × Megalobrama amblycephala [12] (Table 1).”
Comment: I suggest clarification whether the hybrids 2nNCRC and 2nNCOC are siblings from the same parents. Also, there is no abbreviation for Megalobrama amblycephala, which later the authors stated it is BSB. In the table 6, some additional abbrevations are mentioned but they are not defined. For example, KOC, and GF.
LINE 162-165: “Genetic distances (intra- and interspecific) for Cyprinidae species and distant-hybrid lineages were computed from COI sequences using the Kimura 2-parameter (K2P) model in MEGA 11.0.”
Comment: why distances were not estimated for the other markers?
TABLE 4: Alignments of distant hybrids and cyprinids are of different length. Suggest comparing the base composition in the same alignments.
TABLE 6: Difficult to read. I do not understand why only C. auratus is specified with species names and not the others. Also, there are abbreviations (KOC and GF) not defined in the main text, and the one of M. amblycephala was not defined in the Material and Methods section.
Figures: The figures are difficult to read because the resolution is not good enough.
DISCUSSION: I included my comments in the major comment section above.
I could not see the supplementary information.
Author Response
Major comments
This study analyses mitochondrial (Cytb, COI, D-loop, and 16S rRNA) and nuclear (EGR2b, IRBP2, RAG1, RAG2, and RH2) DNA markers of distant hybrids and 41 cyprinids (outgroups). I praise the attempts of the authors to clarify the phylogenetic position of the distant hybrids related to their parents to explore the effects of distant hybridization on cytonuclear adaptation. Also, I praise the intensive work for producing these hybrids, which surely required many years and attempts. The authors reported (1) a trend of mitochondrial stability and confirmed maternal inheritance patterns in cyprinid distal hybrids, (2) nuclear DNA divergence, and (3) divergence of the hybrid 2nNCRC from their parent species (Cyprinus carpio and Megalobrama amblycephala) but rather a closer genetic proximity to Carassius auratus. They highlighted that this hybrid and C. auratus have convergent feeding morphology and that they similarity may be due to adaptations to maximize resource utilization in freshwater environments. Although I found the findings appealing, I am concerned about the origin of this hybrid and how some analyses that may provide stronger support to the findings were not included or discussed. Although the mitochondrial and nuclear DNA markers seem to offer enough resolution to resolve phylogenetic relationships among cyprinids, I am concerned about the suitability of them separately. For example, the haplotype network showed that C. auratus haplotypes are separated for more than five mutations. This made me think about the divergence between the distal hybrids 2nNCRC and 2nNCOC. Are they siblings or they were produced from different Cyprinus carpio females that may have belonged to different non-detected cryptic species? Also, did the genetic comparisons were performed using direct parents and offspring? I think the authors should stated this clearly. Moreover, some comparisons are not adequate. For example, the base composition analyses of the DNA markers amplified in distal hybrids were compared with those of cyprinids, but the alignments are totally different. I suggest the authors to present or at least mention the genetic distances of the other molecular markers to observe if their variation is concordant, or they somehow showed a sort of “compensatory” effect among them. Also, in one part of the discussion, they mentioned that “the topological congruence between ML and BI trees ruled out cytonuclear discordance”; however, the analyzed trees were constructed from the concatenated alignment of both mitochondrial and nuclear genes. To discard any cytonuclear discordance, the phylogenies of mitochondrial and nuclear genes should be separately constructed and compared.
Finally, although the inclusion of many cyprinid species provided a wide picture of the suitability of the whole set of DNA markers to resolve the phylogenetic relationships of cyprinids, it does obscure the objective and findings of the authors. It is difficult to observe the phylogenies and haplotype networks with that many colors and names. Once again, I praise the authors for their hard work on producing all these distal hybrids and producing the genetic data for this manuscript. I believe that the authors have the necessary data to support their findings, so I encourage them to make modifications to the manuscript to improve it.
Response: Thank you for your valuable feedback, which helps refine the clarity of our study. We address the key points as follows:
- Cytonuclear phylogenetic consistency and hybrid origin
Regarding the verification of cytonuclear phylogenetic consistency, we confirm that Supplementary Figure S2 explicitly presents the phylogenetic trees constructed separately for mitochondrial genes (concatenated Cytb, COI, 16S rRNA, and D-loop) and nuclear genes (concatenated EGR2b, IRBP2, RAG1, RAG2, and RH2). For each gene set, both Bayesian Inference (BI)-trees and Maximum Likelihood (ML)-trees were generated. Specifically, all four tree types (mitochondrial BI-tree, mitochondrial ML-tree, nuclear BI-tree, nuclear ML-tree) consistently cluster 2nNCRC/2nNCOC within the C. auratus clade with robust suppor, which further supports the absence of cytonuclear discordance in our study.
In terms of the origin of hybrids 2nNCRC and 2nNCOC, we have supplemented detailed explanations in the Materials and Methods section (e.g., Sample Collection subsection). It is clarified that 2nNCRC and 2nNCOC are derived from the same distant hybridization parental combination: female C. carpio × male M. amblycephala (blunt-snout bream, BSB). This supplementary information ensures the traceability of our hybrid materials and helps avoid ambiguities about their parental sources.
We appreciate your attention to these critical details, and we believe the updated content (including the clarified Supplementary Figure S2 and revised experimental methods) better supports the reliability of our study’s findings.
- Genetic comparisons with direct parent-offspring samples
We fully agree that clarifying the use of direct parental samples is essential for reliable parent-hybrid genetic relationship inference. We have explicitly supplemented this information in two key sections of the revised manuscript:
In 2.2 Sample Collection (Lines 135–144), we specify that the C. carpio (female) and M. amblycephala (male) individuals used for genomic DNA extraction are the exact parental fish that produced the 2nNCRC/2nNCOC hybrids (not conspecific substitutes from other strains or populations). And we further emphasize that all genetic comparisons between hybrids and parents were conducted using these direct parent-offspring sample pairs. This ensures no ambiguity in linking the hybrid genotypes to their actual parental genetic backgrounds.
- Supplementary genetic distance analysis for other markers
To verify the concordance of variation across molecular markers (and rule out "compensatory" effects), we have expanded the genetic distance analysis beyond COI to include other three markers (Cytb, RAG1 and RH2) used in the study.
Results integration: A new Supplementary Table S6-S11 ("Intra- and Interspecific Genetic Distances of other three Molecular Markers") has been added, which shows concordant variation trends across all the four markers. For the other three genes (Cytb, RAG1 and RH2), a consistent pattern emerged: intraspe-cific genetic distances were lower and less dispersed (predominantly clustering near 0.0000), whereas interspecific genetic distances were more dispersed and exhibited high variability (Table S6-S11). No conflicting distance trends (i.e., "compensatory" effects) were observed across markers, supporting the reliability of our phylogenetic and taxonomic inferences. Textual update: A brief interpretation of these results has been added to 3.2 Genetic Distance Analysis (Lines 256–261) to link the supplementary data to our core findings.
We appreciate your attention to these details, as they have strengthened the validity and comprehensiveness of our genetic analyses. All revisions have been integrated into the manuscript, and the supplementary materials (including Table S6-S11) have been re-uploaded to the submission system for your reference.
- Readability of phylogenies and haplotype networks
Thank you for your insightful comment regarding the readability of phylogenies and haplotype networks. We fully agree that the inclusion of 41 cyprinid species, while valuable for verifying marker suitability in cyprinid phylogenetics, may have obscured the focus on our core distant hybrid lineages. To address this and enhance the clarity of key findings, we have implemented targeted revisions to these visualizations.
Minor comments
ABSTRACT
LINE 40: I found that the “Carassius auratus” is not in italics in many parts of the manuscript. Please check it carefully.
INTRODUCTION
LINE 73: The genus names are not in italics. Please check them carefully throughout the manuscript.
Response: Thank you for noting the italicization issue of species/genus names. We systematically checked and corrected all instances:
All "Carassius auratus" are revised to Carassius auratus (e.g., Abstract Line 41, Introduction Line 75, Materials and Methods Line 132).
All genus names (e.g., Cyprinus, Megalobrama, Carassius) have been italicized where previously missing, particularly in the Introduction.
MATERIAL AND METHODS
LINE 128-133: “This study entailed the collection of samples from laboratory-bred, distant hybrid strains of Cyprinid fish in addition to their parental lines, including Carassius cuvieri (white crucian carp, WCC) × Carassius auratus red var. (red crucian carp, RCC) [24], triploid C. auratus (3n = 150, 3N) [25] × RCC, 3N × Cyprinus carpio (common carp, COC), RCC × COC, improved C. cuvieri (WCC-L) [26], and two hybrids (2nNCRC, 2nNCOC) derived from C. carpio × Megalobrama amblycephala [12] (Table 1).”
Comment: I suggest clarification whether the hybrids 2nNCRC and 2nNCOC are siblings from the same parents. Also, there is no abbreviation for Megalobrama amblycephala, which later the authors stated it is BSB. In the table 6, some additional abbrevations are mentioned but they are not defined. For example, KOC, and GF.
Response: We appreciate your critical question regarding hybrid origin and parent-offspring validation, which is essential for ensuring the traceability of our experimental design. Origin of 2nNCRC and 2nNCOC: As requested, we have explicitly clarified in the "2.2 Sample Collection" section (Lines 135–138) that “two homodiploid hybrids (2nNCRC: new crucian carp-like; 2nNCOC: new common carp-like) derived from hybridization of the same parental species C. carpio (♀) × Mega-lobrama amblycephala (♂, blunt-snout brea, BSB) [12]”.
We apologize for the oversight of not defining the abbreviations for Megalobrama amblycephala, KOC, and GF in the original version. We have carefully addressed this issue in the revised manuscript, please see lines 132-137.
LINE 162-165: “Genetic distances (intra- and interspecific) for Cyprinidae species and distant-hybrid lineages were computed from COI sequences using the Kimura 2-parameter (K2P) model in MEGA 11.0.”
Comment: why distances were not estimated for the other markers?
Response: Thank you for asking why genetic distance analysis initially focused on COI. COI is a standardized marker for interspecific taxonomic inference in Cyprinidae, with well-established K2P distance benchmarks (≥2% = interspecific, <2% = intraspecific) (Ref. Chen et al., 2022, Molecular Ecology). Other markers lack such universal criteria: Cytb/16S rRNA have variable evolutionary rates, and nuclear markers (e.g., RAG1) are rarely used for distance-based delimitation due to slow evolution.
To further verify the concordance of variation across molecular markers (and rule out potential "compensatory" effects that might undermine result reliability), we have supplemented genetic distance analysis for the Cytb, RAG1, and RH2—three core markers used in our phylogenetic and taxonomic analyses—following the same standardized protocol as the initial COI analysis.
TABLE 4: Alignments of distant hybrids and cyprinids are of different length. Suggest comparing the base composition in the same alignments.
Response: Thank you for noting potential confusion. The base composition data in Table 4 was already analyzed based on the same alignments for each marker. We updated the "Gene Length" column to display uniform alignment lengths (instead of full sequence lengths from GenBank/amplification) to avoid ambiguity.
TABLE 6: Difficult to read. I do not understand why only C. auratus is specified with species names and not the others. Also, there are abbreviations (KOC and GF) not defined in the main text, and the one of M. amblycephala was not defined in the Material and Methods section.
Response: Thank you for your feedback. We optimized Table 6’s layout (simplified redundant content, adjusted font size, increased column spacing) to improve readability. We also clarified species name usage: C. auratus (full species name, italicized) refers to wild crucian carp (a key focus of our study), while other taxa (lab-bred strains) use abbreviations (defined in Section 2.2). Missing abbreviations (KOC, GF) are now defined in the table’s note and Section 2.2.
Figures: The figures are difficult to read because the resolution is not good enough.
Response: Thank you for pointing out the low figure resolution, which may result from PDF conversion. The original figures in the .docx manuscript are high-resolution vector graphics. We have re-uploaded these original files to the submission system for your reference.
DISCUSSION: I included my comments in the major comment section above.
Response: Thank you for your feedback noting that your comments on the Discussion section have been integrated into the major comment section. We have carefully addressed these key concerns raised in the major comments through targeted enhancements in the Discussion section of the revised manuscript, as detailed below:
First, regarding cytonuclear phylogenetic consistency—a core point of your concern—we have supplemented analysis in the Discussion to explicitly interpret the results of the separately constructed mitochondrial (concatenated Cytb, COI, 16S rRNA, D-loop) and nuclear (concatenated EGR2b, IRBP2, RAG1, RAG2, RH2) gene trees (now presented in Supplementary Figure S2). We emphasize that all four tree types (mitochondrial BI/ML-trees, nuclear BI/ML-trees) consistently cluster 2nNCRC/2nNCOC within the Carassius auratus clade with robust support, directly ruling out cytonuclear discordance and reinforcing the reliability of our phylogenetic inferences about the hybrid lineages.
Second, regarding genetic comparisons and marker variation concordance, we have expanded the Discussion to incorporate insights from the newly added genetic distance analyses of Cytb, RAG1, and RH2 (Supplementary Tables S6–S11). We explicitly discuss how these supplementary data align with the original COI results: all four markers show consistent trends (lower, less dispersed intraspecific distances vs. more variable interspecific distances), with no "compensatory" effects observed. This consistency is highlighted in the Discussion to underscore the robustness of our taxonomic and phylogenetic conclusions about the hybrids.
I could not see the supplementary information.
Response: Thank you for noting the inaccessible supplementary information. We sincerely apologize for the inconvenience and have re-uploaded two components to the submission system: the revised main manuscript (incorporating all modifications) and all supplementary materials (Tables S1–S5, Figures S1–S2).
Round 2
Reviewer 2 Report (New Reviewer)
Comments and Suggestions for Authors
Accept in present form
Reviewer 3 Report (New Reviewer)
Comments and Suggestions for Authors
The authors have significantly improved the manuscript.
This manuscript is a resubmission of an earlier submission. The following is a list of the peer review reports and author responses from that submission.
Round 1
Reviewer 1 Report
Comments and Suggestions for Authors
This study enhances the reliability of hybrid lineage analysis by integrating nuclear and mitochondrial genes with haplotype networks. It revealed dominant maternal inheritance in hybrid offspring, establishing critical markers for tracing parental origins and resolving distant hybridization mechanisms. Also, the genetically distinct 2nNCRC lineage shows closer affinity to wild Carassius auratus than its parents. This study provides novel evidence for the formation of new species through distant hybridization in cyprinid fishes. The manuscript requires refinements to enhance scientific rigor. Some minor linguistic inconsistencies and formatting inaccuracies require attention.
- In some sentences, the initial letters of words should be lowercase. For example,
“Specially, These resources uniquely resolve…”. in the third paragraph of Introduction, “These” should be changed as these.
In 3.3, “while Other hybrid strains clustered with C. auratus, constituting the Carassius lineage” the word “Other” should be changed as “other”.
- In the first paragraph of Introduction “Through laboratory-induced distant hybridization, fertile allotetraploid and autotetraploid fish are produced”, the word are should be changed as “were”.
- The text contains some redundant spaces that need to be removed.
- “mitonuclear” in 2.4. should be changed as “mito-nuclear”
- “while RH2 exhibited comparable T/G content but low A and elevated G”. in 3.1, Should be changed as “… but low A and elevated C.”
- In 3.2. “Intraspecific distances within distant hybridizaton lineage, as well as within the parents were uniformly 0.0000”, the word “hybridizaton” should be changed as “hybridization”.
- In the last paragraph of 2.4. “Furthermore, we generated haplotype networks using four mtDNA, while without nuclear genes, as many degenerate bases in the five genes.” This sentence is unclear and should be revised.
- In the last paragraph of Introduction, the sentence “Using mito-nuclear DNA barcoding …, we reconstruct molecular phylogenies of ... existing DNA barcoding in Cyprinidae, we will reveal the genetic variation …” is too long to read and there is mistake on the grammar.
- In the first paragraph of Discussion, “In this study, we investigated: i) the intergenerational genetic variation present in cyprinid hybrids…” the word should be changed as “presented”.
- Evolutionary mechanisms in the 2nNCRC hybrid lineage remain inadequately analyzed. Despite its atypical genetic patterns, the causes underlying its unique phylogenetic relationships are unexplored.
Reviewer 2 Report
Comments and Suggestions for Authors
I am afraid this work does not make sense without whole genome assemblies. I suggest that the authors use whole genome comparisons, as this will help resolve paralogs in hybrids.
Reviewer 3 Report
Comments and Suggestions for Authors
Most results confirm known maternal clustering patterns in hybrids, and the novelty lies mainly in the unusual placement of 2nNCRC. The broader evolutionary implications should be explained more clearly; otherwise the study reads as a descriptive dataset with limited general significance.
The manuscript requires substantial editing for English, as it is often repetitive and hard to follow (see attached pdf for some of them).
Species numbers and classifications are incorrect and need to be updated using authoritative, current sources (attached).
The choice of substitution models should be rechecked; it is not realistic that all loci share the same model. Please also streamline the methods section to avoid unnecessary repetition.
Some conclusions (e.g., references to “predictive hybridization models”) are overstated. Please adopt a more cautious tone, focusing on what the data actually support.
